# New flow relaxation mechanism explains scour fields at the end of submarine channels

F. Pohl [1]*, J.T. Eggenhuisen [1], M. Tilston[1] & M.J.B. Cartigny [2]

Particle-laden gravity flows, called turbidity currents, flow through river-like channels across the ocean floor. These submarine channels funnel sediment, nutrients, pollutants and organic carbon into ocean basins and can extend for over 1000's of kilometers. Upon reaching the end of these channels, flows lose their confinement, decelerate, and deposit their sediment load; this is what we read in textbooks. However, sea floor observations have shown the opposite: turbidity currents tend to erode the seafloor upon losing confinement. Here we use a state-of-the-art scaling method to produce the first experimental turbidity currents that erode upon leaving a channel. The experiments reveal a novel flow mechanism, here called flow relaxation, that explains this erosion. Flow relaxation is rapid flow deformation resulting from the loss of confinement, which enhances basal shearing of the turbidity current and leads to scouring. This flow mechanism plays a key role in the propagation of submarine channel systems.

---

[1] Faculty of Geosciences, Utrecht University, P.O. box 80021, 3508 TA Utrecht, The Netherlands. [2] Department of Geography, Durham University, Lower Mountjoy South Road, DH1 3LE Durham, UK. *email: florian.pohl63@gmail.com

Turbidity currents are particle-laden gravity flows that move downslope because of the density difference between the sediment-laden flow and the ambient water. They represent a major transport agent for sediment in the ocean, and the associated turbidite deposits are a sink for organic carbon burial[1,2], host for major reservoirs of hydrocarbons[3], and are also a potential depot for plastic debris[4]. On the ocean floor, turbidity currents typically transport sediment within confinements such as channels, which focus the flow and predominantly prevent deposition of the suspended sediment[5]. Upon leaving the channels, turbidity currents lose their lateral confinement and deposit their sediment load in lobate sediment bodies, thereby forming some of the largest sediment accumulations on Earth[6]. While sediment transport in channels and deposition on lobes is reasonably well understood, it is not clear why these two systems are connected by a transition zone characterized by enhanced erosion, referred to as the channel-lobe transition zone (CLTZ) (Fig. 1a)[7].

It is surprising that the area downstream of a channel is marked by erosion as the lateral expansion and associated deceleration of turbidity currents upon leaving the channel would suggest deposition. Previous research has established that a turbidity current leaving channel confinement spreads laterally, and that lateral spreading increases the overall friction of the flow, resulting in deceleration and deposition of suspended sediment[8,9]. Yet bathymetric surveys on modern CLTZs show repetitive erosive structures, so-called scour fields, instead of the anticipated deposits (Fig. 1a)[10–15]. These scour fields can be >100 km long with individual scours up to 20 m deep and 2500 m long[11,13]. Erosion of the ocean floor at the CLTZ inherently plays a critical role in the development and the propagation of channel systems (Fig. 1b)[16–20]. Although erosive features of CLTZs are well documented, the dominant conceptual model to explain their genesis remains speculative and has not been subjected to rigorous experimental evaluation.

The favored hypothesis within the literature to explain erosion at channel terminations is the occurrence of a hydraulic jump (i.e., the transition from Froude supercritical to Froude subcritical flow) as the turbidity current leaves the lateral confinement[7,13,21]. A hydraulic jump is expected to increase the erosion potential of the flow, as turbulence and bed shear stresses are increased locally[21–24]. However, there is no study that confirms the link between erosion processes and hydraulic jumps.

Here we use the newly developed Shields scaling approach[25] to directly observe the pattern of erosion and deposition resulting from a turbidity current leaving lateral confinement in a flume set-up (Fig. 2a). Additionally, we conduct a reference experiment in which the flow remains confined over the entire slope (Fig. 2b). This experimental method allows us to observe the dynamic interaction between the turbidity current and the sea floor in relation to the loss of confinement. The observed incision at the CLTZ is explained by a flow mechanism which we term flow relaxation. Flow relaxation results from the loss of lateral support to the turbidity current by the channel walls and plays a crucial role in channels propagating across the ocean floor.

## Results

**Flume experiments.** The experimental results show the anticipated enhanced erosion downstream of the loss of confinement (Fig. 3a). Upstream of the loss of confinement both experiments displayed the expected similar behavior (Fig. 3a–c). Upon losing confinement, however, the unconfined flow incised deeper along the down-flow trajectory than in the reference experiment without the loss of confinement. The incision in the center was flanked by deposition of a ~2 cm high ridge on each side (Fig. 3a). In contrast, the reference experiment showed less incision and no depositional ridges (Fig. 3b). The resulting incision flanked by depositional ridges resulted in an extension of the confinement.

The downstream development of the self-confinement over time suggests a link between the enhanced erosion and incipient channel development. The propagation of the confinement was captured by the velocity probes (Fig. 3d). The enhanced erosion (i.e. decrease in bed elevation) stopped just downstream of the loss of confinement (at UVP 4) after ~40 s of the experiment (Fig. 3d), when the new self-confinement was established. Further downstream, at UVP 5, erosion occurred over a longer time period of ~80 s (Fig. 3d), implying a later establishment of self-confinement at this more distal location. Hence, the establishment and the propagation of the self-confinement in the experiment was driven by on-axis erosion and off-axis deposition downstream of the loss of confinement.

The turbidity current immediately spread upon leaving the confinement. This spreading lowered the elevation of the velocity maximum, and resulted in an increased basal shear stress, which enhanced the erosion potential of the flow. The velocity of the

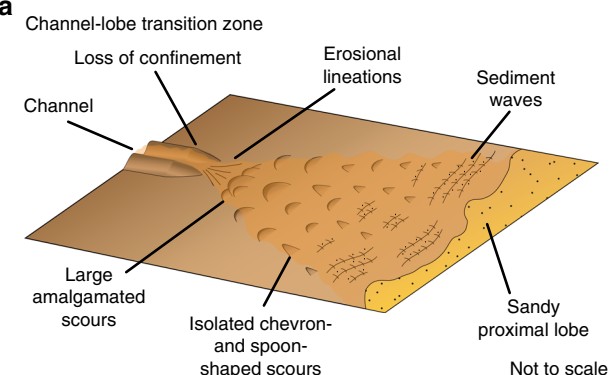

**a**

Channel-lobe transition zone

Loss of confinement

Erosional lineations

Sediment waves

Channel

Large amalgamated scours

Isolated chevron- and spoon-shaped scours

Sandy proximal lobe

Not to scale

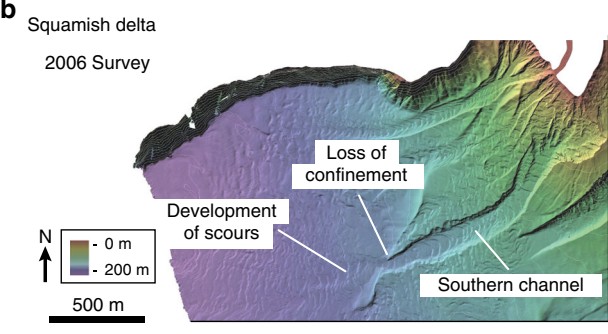

**b** Squamish delta

2006 Survey

Loss of confinement

Development of scours

Southern channel

N

- 0 m
- 200 m

500 m

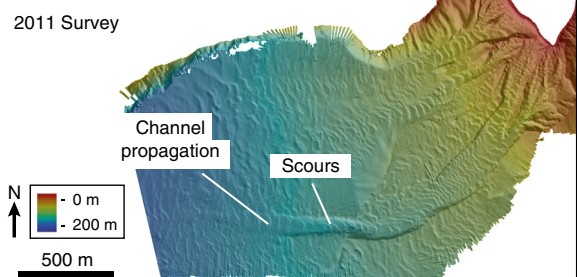

2011 Survey

Channel propagation

Scours

N

- 0 m
- 200 m

500 m

**Fig. 1** Examples of systems with a loss of confinement. **a** Sketch of a channel-lobe transition zone based on bathymetrical surveys. Modified from ref. [13]. **b** The Squamish Delta. In the upper bathymetry map, the channel termination is marked by a rapid loss of confinement[43]. The lower bathymetry map was obtained 5 years later and shows how erosion has led the propagation of the channel by ~400 m[41]

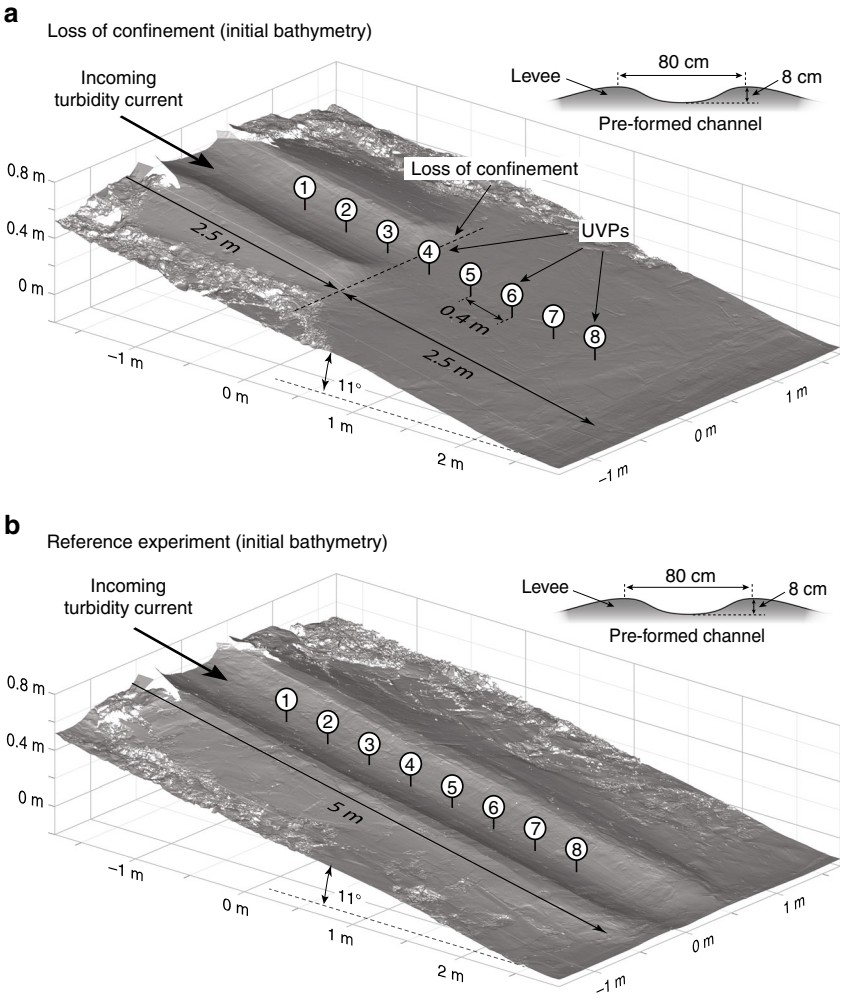

**Fig. 2** Digital-elevation-models of the initial bathymetry. **a** The flume experiment with loss of confinement. The loss of confinement was generated by a decrease of the levee height 2.5 m downstream of the inlet box. **b** The reference flume experiment with a continuous pre-formed channel over the entire length of the slope. The substrate in both experiments was equivalent to that of the sediment mixture used to generate the turbidity currents. The channel dimensions, as well as the input conditions of the incoming turbidity current, were identical in both experiments. UVP Ultrasonic Velocimetry Probe

turbidity current was captured by 8 velocity probes aligned along the channel thalweg (Fig. 2a, b). Each of the probes collected a full vertical velocity profile of the flow (Fig. 4a, b). The turbidity current accelerated down the channel as it entered the setup. Downstream of the loss of confinement the flow decelerated (Fig. 4c). Deceleration was accompanied with a decrease in flow thickness due to lateral spreading upon leaving the channel (Fig. 4d). However, due to the lowering of the velocity maximum, the velocity gradient at the flow base was increased (cf. Fig. 4a), which enhanced the friction between the flow and the bed, i.e., the bed shear velocity (Fig. 4e). This increased shear velocity was responsible for the enhanced erosion downstream of the loss of confinement.

## Discussion

The Shields scaling approach allows us to link the morphological changes at the loss of confinement to rapid flow deformation, and its associated enhanced erosion. Our results indicate that this deformation consists of a lowering of the velocity maximum associated with lateral spreading of the flow field. The mechanism leading to this transformation is explained through the concept of flow relaxation, which describes the reaction of the flow to the development of strong lateral pressure gradients upon exiting the channel (Fig. 5a, b).

We propose that changes in the lateral pressure gradient at the base of the flow explain the concept of flow relaxation. Within turbidity currents, hydrostatic pressure is increased by the mass of the overlying suspended particles, and since particle concentration decreases from the bed to the top of the current, so does the pressure[26]. The lateral pressure gradient is zero in a channelized flow, due to the absence of horizontal density gradients (Fig. 5a)[27]. When the flow loses confinement, a lateral pressure gradient develops as the turbidity current is denser than adjacent ambient fluid (Fig. 5b). This lateral pressure gradient is strongest at the bottom of the current, which explains the rapid basal evacuation and the lowering of the high velocity core. It is the lowering of this high velocity core that leads to an increase of the near-bed velocity gradient and bed shear velocity (Fig. 4a, d, e), triggering scour development. In this model, the area between the proximal and distal regions of the scour field is interpreted as the distance over which the current re-equilibrates to the new unconfined flow conditions. In summary, rapid flow deformation and associated scour formation that occurs over this re-adjustment range is explained through changes in lateral pressure gradients as illustrated in the flow relaxation model (Fig. 5a, b).

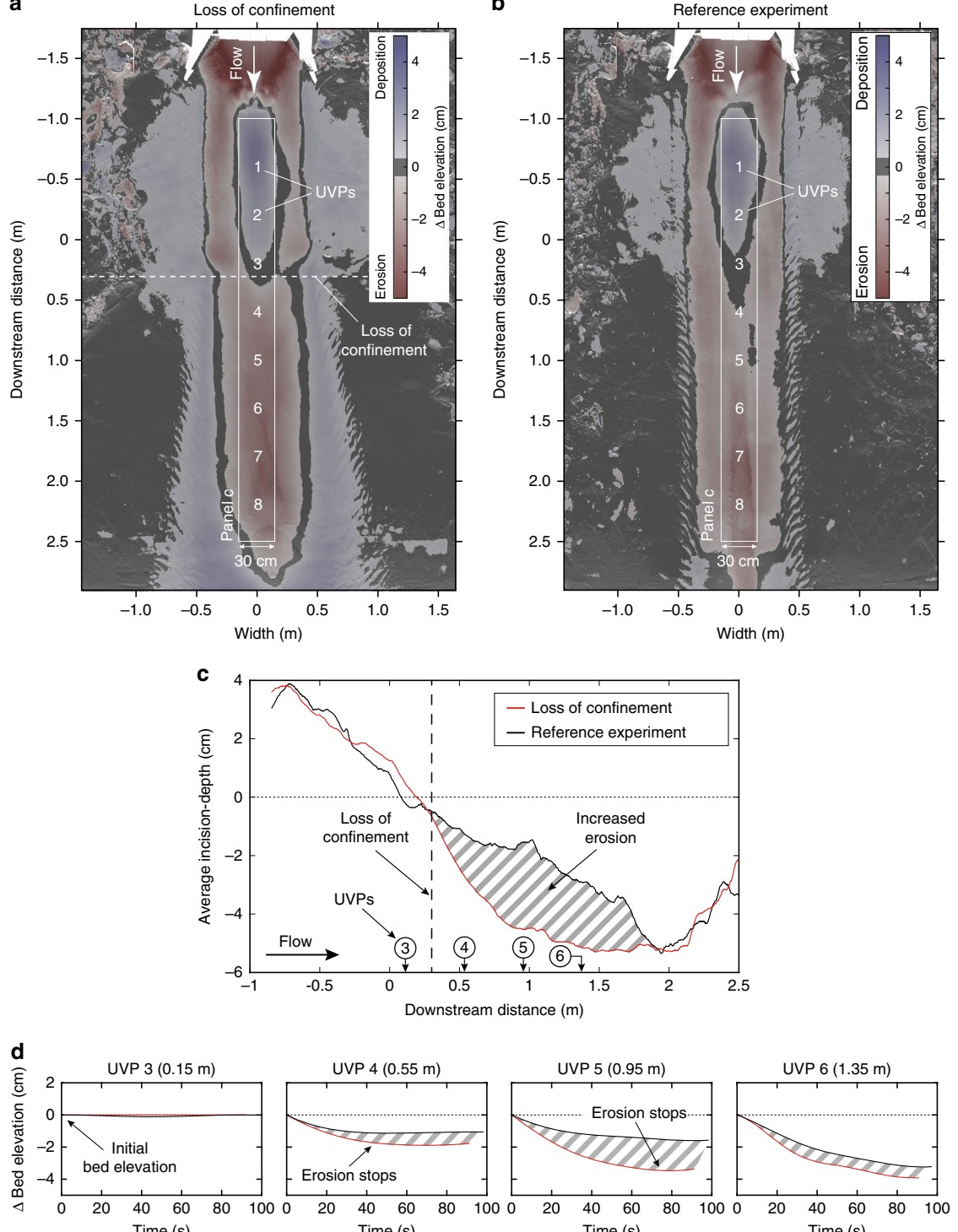

**Fig. 3** Erosion and deposition in the two experiments. **a** Map showing erosion and deposition in the experiment with the loss of confinement. **b** Erosion and deposition in the reference experiment. **c** Laterally averaged incision-depth of a 30 cm wide strip along the channel thalweg. Erosion in the experiment increased with the loss of confinement. **d** Bed-elevation change during the experiments as captured in the footprint of the Ultrasonic Velocimetry Probes (UVP) (see Methods). Bed-elevation change was generally higher in the experiment with the loss of confinement than in the reference experiment. The difference was highest below UVP 5, which was located 0.95 m downstream of the loss of confinement

Research to date has tended to ascribe the formation of scour fields in CLTZs to hydraulic jumps[7,13,14,21,28,29] which cause enhanced turbulence and consequently, increased erosion[21–23]. In our experiments, we did not observe a hydraulic jump as the flow thickness did not increase upon leaving the confinement (Fig. 4d), while a hydraulic jump would result in thickening of the flow[29].

Previous experiments in saline density flows without suspended particles have observed a hydraulic jump at the channel termination[20]. However, this hydraulic jump was correlated with late-stage topographic forcing through channel mouth bar development, rather than the loss of confinement. Hydraulic jumps downstream of a loss of confinement have been observed in

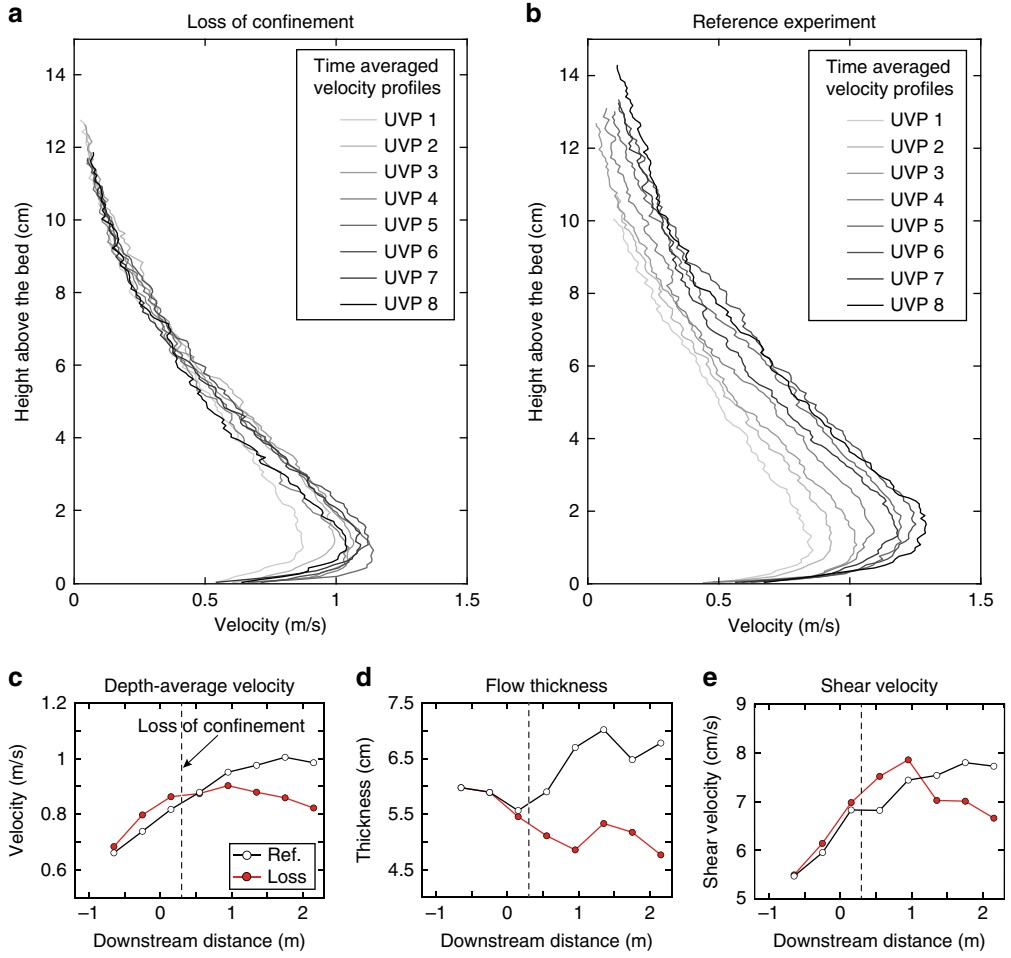

**Fig. 4** Flow-dynamic parameters captured by the velocity probes. **a** Time-averaged velocity profiles for the turbidity current in the experiment with loss of confinement. **b** Time-averaged velocity profiles for the turbidity current in the reference experiment (see Figs. 2 and 3 for probe locations). **c** Depth-averaged velocity. The turbidity current downstream of the loss of confinement was slower than the turbidity current in the reference experiment. **d** Flow thickness. After leaving the confinement the calculated turbidity current thickness decreased. **e** Shear velocity. Shear velocity downstream of the loss of confinement was increased in comparison with the reference experiment. See Methods for details on average calculations

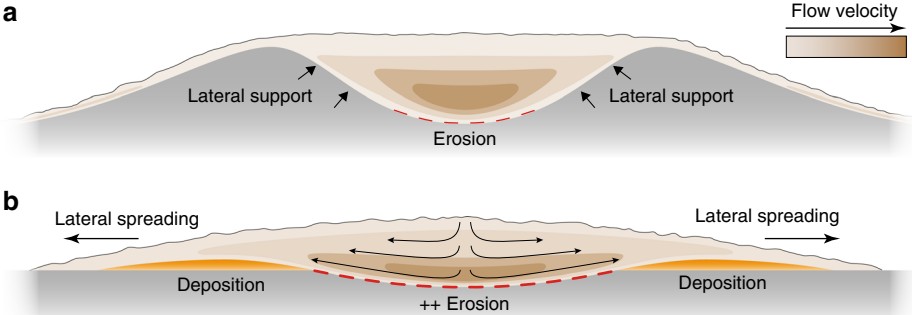

**Fig. 5** Illustration of the flow relaxation model. **a** Flow confined in a channel. The channel side-walls counteract the near-bed lateral pressure differences within the flow, resulting in a lateral pressure gradient of zero. Note lateral and vertical variations in the flow velocity field after[27]. **b** A flow that relaxes upon leaving a confinement. The loss of lateral support by the channel flanks results in a lateral pressure gradient within the flow, and hence, lateral spreading triggers a lowering of the velocity maximum. This shifts the height of the maximum velocity bed-wards and increases the shear stress at the bed, resulting in erosion. Lateral to the incision, levee-shaped sediment bodies are deposited due to the lateral decrease in flow velocity

particle-laden density flows (turbidity currents) in experiments by Baas et al.[30]. However, the loss of confinement in those experiments was accompanied with a sudden decrease in slope gradient. Hydraulic jumps are known to frequently occur due to sudden decreases of the slope[29,31,32]. Therefore, it seems more likely that the hydraulic jump in the experiments by Baas et al.[30] was

triggered by the break in slope rather than the loss of confinement. Moreover, a single hydraulic jump forms a single scour rather than scour fields, as observed in CLTZs (Fig. 1a, c). Monitoring of saline flows in the Black Sea channel has revealed that each scour is associated with an individual hydraulic jump[14,33]. Consequently, Dorrell et al.[14] have proposed the

presence of a hydraulic-jump-array associated with the formation of a scour field in CLTZs. However, the density structure of the Black Sea saline flows is different from the density structure of a turbidity current[34–36], and therefore it remains questionable whether such a hydraulic-jump-array model translates across to turbidity currents. Furthermore, the hydraulic jumps in the Black Sea formed within the confinement of a channel, rather than at the loss of confinement characteristic of CLTZs[14,33]. Finally, a third existing model explains multiple scours by the impingement of vortices on to the ocean floor beneath a hydraulic jump[24]. In this model, each individual impingement would form a scour. However, scour formation by impingement of vortices has never been replicated in experiments. Overall, the association of scour fields in CLTZs with hydraulic jumps remains open for debate.

Flow relaxation provides a novel mechanism to explain the formation of scour fields in CLTZs. Instead of going through a hydraulic jump, the flow relaxes upon leaving the confinement, which enhances the basal shear stress of the turbidity current (Fig. 4a, e). This increase of the erosional potential also increases the potential for scour formation over the entire area in which the flow relaxes. Experiments have shown that erosive conditions can trigger the formation of fields of individual scours, in which the location of individual scours is controlled by random irregularities on the ocean floor[37,38]. Such irregularities could consist of patches of coarser grains, a series of small-scale bedforms, a scour surrounding a boulder, or any biological feature. The formation of individual scours at these irregularities would then result in the formation of scour fields.

Submarine channels can grow to extraordinary lengths, like the Northwest Atlantic Channel, which extends ~3800 km[39]. Additionally, these submarine channels can propagate at exceptional rates of up to ~500 m yr$^{-1}$ in the Amazon system[40] or ~80 m yr$^{-1}$ in the much smaller Squamish Delta[41]. These high rates suggest the existence of a very efficient channel propagation mechanism. The nature of channel propagation mechanisms is much debated, where attention has so far focused on whether the propagation of submarine channels is dominantly due to erosion or deposition[16–19,25,42]. Hamilton's et al.[20] experimental saline density flows show an increase in the sediment transport capacity of the flow at the channel mouth, and they proposed erosion as the impetus for sustained channel propagation. Our results provide the physical processes that drive this erosion and demonstrate the applicability of the processes in sediment-laden flows, such as turbidity currents. As the flow relaxes at the channel termination, it incises in the center and deposits levee-shaped sediment bodies off-axis to both sides, efficiently forming a self-confinement (Figs. 3a and 5b). The increased levels of self-confinement start to provide lateral support to the flow, which results in a decrease of the lateral pressure gradient, and prevent spreading of the lower part of the flow. Hence, the self-confinement is damping the effect of the flow relaxation and thus the erosion potential of the flow. Self-confinement establishes until an equilibrium channel shape is reached, thereby extending the channel further across the ocean floor.

Channel propagation in the experiment resulted in a more gradual loss of confinement, which still triggered flow relaxation and erosion. Additionally, channel propagation in natural systems will depend on the size of the turbidity currents with respect to the channel dimensions. If turbidity currents are too small to reach the end of the channel, flow relaxation cannot occur and the channel will not propagate, as seen for the Southern Channel in the Squamish Delta between 2004 and 2006[43,44]. On the other hand, if a turbidity current is too large for the channel, then the flow will overspill the channel, thereby reducing its size until the flow matches the channel dimensions[45], in which case flow relaxation will occur.

Our model provides a mechanism to explain the propagation of a channel in the Squamish Delta. A bathymetric survey of the Squamish Delta, that was conducted in 2006, showed that the Southern Channel terminated with a rapid loss of confinement (Fig. 1b)[43]. A subsequent bathymetry survey in 2011 revealed propagation of the Southern Channel over a distance of ~400 m (Fig. 1b)[41]. Channel propagation was generated by the incision into the underlying substrate downstream of the rapid loss of confinement as demonstrated by 93 daily repeat surveys performed in 2011[46]. Hence, propagation of the Southern Channel was driven by erosion comparable to our experiments (Fig. 3a).

In summary, our results provide measurements of a turbidity current that enhances its erosion potential after leaving a channel. Upon leaving the channel confinement the turbidity current spreads laterally, bringing the velocity maximum closer to the bed, which causes an increase in the bed shear stress and erosion. The here introduced model of flow relaxation provides a flow-dynamic process that is pivotal for the development of scour fields in CLTZs, and plays a central role in the propagation of submarine channels.

## Methods

**Scaling approach.** The turbidity currents were downscaled from natural to experiment size by using Shields scaling[25]. This technique relies on two scaling parameters: (1) The Shields parameter, which is kept close to natural values, and (2), the boundary Reynolds number, which is relaxed as long as rough to transitionally rough boundary layer conditions are maintained (Supplementary Fig. 1). Together, these two parameters predict whether the current will erode or deposit sediments and whether the particles will be transported as bedload or suspended load.

The Shields parameter describes the ratio between the shear stress and the gravity force acting on particles[47]:

$$\theta = \frac{\rho_t u_*^2}{(\rho_s - \rho_w)g d_t}, \tag{1}$$

where $\rho_s$ is the density of the suspended sediment (quartz sand with 2650 kg m$^{-3}$), $\rho_w$ the density of water (1000 kg m$^{-3}$), $d_t$ the grain size of the suspended sediment, $g$ the gravitational force (9.81 m s$^{-2}$), and $u_*$ the shear velocity (Eq. 3). The density of the turbidity current $\rho_t$ is:

$$\rho_t = (\rho_s - \rho_w)C + \rho_w, \tag{2}$$

with $C$ as the sediment concentration at the inlet. Using the inlet concentration throughout the domain is validated by the fact that only ~2% of the initial sediment volume is deposited on the slope. Thus, changes to the amount of sediment in transport due to deposition are minor. The shear velocity $u_*$ can be derived from the shape of the velocity profile below the velocity maximum $U_{max}$, by assuming a logarithmic velocity profile between the bed and the height of the velocity maximum $h_m$[25,48–50]:

$$u_* = U_{max}\kappa\left(\ln\left(\frac{h_m}{0.1d_{90}}\right)\right)^{-1}, \tag{3}$$

where $\kappa$ is the von Kármán constant with a value of ~0.4. The $d_{90}$ is derived from the grain-size distribution in the turbidity current. Studies of natural turbidity currents revealed a typical value for the Shields parameter of 1–10 (Supplementary Fig. 1)[2,51]. In our experiments, we meet these values by varying the sediment concentration, and adjusting the velocity of the flow by varying the slope accordingly.

The boundary Reynolds number $Re_p$ controls the hydraulic conditions of the viscous sub-layer, from hydraulically smooth ($Re_p < 5$), to transitional ($5 < Re_p < 70$), to hydraulically rough ($Re_p > 70$)[52]. In the hydraulically rough regime, the viscous sub-layer is dominated by turbulent forces, whereas in a hydraulically smooth regime the viscous sub-layer is dominated by viscous forces. Studies report a transitionally rough regime for natural turbidity currents (Supplementary Fig. 1)[2,53]. The value of the $Re_p$ is given by the ratio of the grain size to the thickness of the viscous sub-layer:

$$Re_p = \frac{u_* d_b}{\nu}, \tag{4}$$

where $d_b$ is the grain size of the sediment of the bed, and $\nu$ is the kinematic viscosity of clear water at 20 °C ($1 \times 10^{-6}$ m$^2$ s$^{-1}$). In the experiments, we meet the transitionally rough hydraulic regime by using a fine grain size ($d_{10} = 35$ μm, $d_{50} = 133$ μm, $d_{90} = 214$ μm) for the sediment of the bed (Supplementary Fig. 2). We also use the same grain size for the suspended sediment of the turbidity current to avoid changes in bed grain size due to deposition from the flow.

**Experiment setup and procedure**. The turbidity currents were released into a 11 m × 1.3 m × 6 m (length × height × width) basin, filled with fresh water (Supplementary Fig. 3). The floor consists of a 5 m long slope of 11°, followed by a horizontal basin floor of 6 × 6 m at the base of slope (Supplementary Fig. 3). The turbidity current was generated from a 0.9 m³ mixture of sediment and water prepared in a separate mixing tank using quartz sand with a mean density of 2650 kg m⁻³, particle diameter ($d_{50}$) 133 μm (Supplementary Fig. 2), and volumetric concentration of 17%. The mixture was pumped into the basin with a radial-flow pump with a constant discharge of 30 m h⁻¹. The discharge was monitored with an electromagnetic flow-meter (Krohne Optiflux 2300) (Supplementary Fig. 4). The turbidity current entered the setup at the upper end of the slope through an inlet box and flowed downslope driven by its excess density.

The initial bathymetry in the experiment consisted of an 11° sloping basin floor with a pre-formed channel that abruptly loses lateral confinement (Fig. 2a). The channel was formed by building confining levees on the slope, and the channel dimensions were 80 × 8 cm (width × depth). Both the levees and the slope were made of loose sand that had the same grain-size distribution as the sand used for the turbidity current (Supplementary Fig. 2). During the experiment, the bulk portion of the flow was contained by the channel, with minimal overspill across the levee crests.

In the reference experiment, a pre-formed channel with identical dimensions was used, but the channel extended over the entire length of the sloping basin floor (Fig. 2b). Besides the difference in channel length, all parameters were kept identical in the two experiments.

**Digital elevation model**. After the release of an experiment current, the basin was drained to expose the deposits. The deposits were scanned by a laser scanner with a measurement accuracy of <0.5 mm. From the laser scan a Digital Elevation Model (DEM) with a horizontal grid spacing of 2 × 2 mm was created. Subtraction of the post-flow DEM from the pre-flow DEM yields a map of the experiment current's deposition and erosion patterns (Fig. 3a, b).

To quantify the erosion during the two runs the average incision-depth was calculated (Fig. 3c). Incision-depth was averaged along the width of a 0.3 m wide corridor, which was aligned within the channel thalweg along the downstream direction. Incision values were laterally averaged to remove local variations in incision depth and therefore represent bulk-averaged trends.

**UVP data acquisition and processing**. An array of eight Ultrasonic Velocimetry Profilers (UVPs) was installed along the channel axis to capture changes in the flow field associated with the abrupt loss of confinement (Fig. 2a, b); UVP acquisition settings are given in Supplementary Table 1. The downstream spacing between individual UVPs was 0.4 m and the probes were set 0.15 m above the bed, facing the upstream direction at an angle of 60° with respect to the basin's initial bed configuration (Supplementary Fig. 5a). Each UVP measures the velocity of sediment grains along the probe's axis, and the bed-parallel velocity component is obtained by trigonometric calculations (Supplementary Fig. 5a); this calculation assumes that the bed-normal component of velocity is zero. Thickness changes of the turbidity current over time suggest a bed-normal velocity component of ~0.01 m s⁻¹. This value is 50 times smaller than the minimum depth-averaged velocity of 0.5 m s⁻¹ and validates the neglection of the bed-normal velocity component. The bed-parallel velocity against time for all UVPs is shown in Supplementary Fig. 6 for experiment with the loss of confinement, and in Supplementary Fig. 7 for the reference experiment. The interface between the flow and sediment bed was discernable as a sharp decrease in velocity (Supplementary Figs. 6 and 7). The vertical bed position was tracked over time, yielding erosion and deposition rates below individual UVPs (Fig. 3d).

Time-averaged velocity profiles were generated over a time interval where the flow was generally steady. The time-averaging interval started 10 sec after the arrival of the turbidity current at the measurement location and was continued over 65 sec (Supplementary Figs. 6 and 7). The time-averaged profiles were then smoothed using a Fourier fitting function to remove spurious spatial velocity fluctuations linked with the UVP's sampling resolution to determine the magnitude $U_{max}$ and the height $h_m$ of the velocity maxima. The flow thickness $h$ is defined here as the height at which the velocity $u$ is half the velocity maximum $U_{max}$ (Supplementary Fig. 5)[54–57]. The depth-averaged velocity was averaged between the bed and the flow thickness $h$.

## Data availability

The datasets presented in the current study are available from the corresponding author upon reasonable request.

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

## Acknowledgements

This contribution is part of EuroSEDS (Eurotank Studies of Experimental Deepwater Sedimentology), supported by the NWO (Netherlands Organization for Scientific Research) (grant no. NWO 864.13.006), ExxonMobil, Shell, and Statoil. We thank Ross Ferguson for proof-reading.

## Author contributions

J.T.E. and M.J.B.C. initiated the EuroSEDS project. F.P. conducted the experiments, analysed the results, drafted the figures, and wrote the initial manuscript. F.P., J.T.E., M.T., and M.J.B.C. contributed to interpretation of the data and writing of the manuscript.

## Competing interests

The authors declare no competing interests.
