## [Peer Review File · Nature Communications]

Reviewers' comments:

Reviewer #1 (Remarks to the Author):

The paper by Pohl and co-authors presents an original mechanism, termed relaxation by the authors, to explain scour field formation at channel-lobe transition zones (CLTZs). Previous works have suggested scour formation at the termination of submarine channels as a result of processes linked to hydraulic jumps. This is an exciting contributing that, in contrast to those previous hypotheses, identifies a new process able to form the scour fields. I would like to congratulate the authors on their clear explanation and supporting figures, and thank them for the really satisfying time I spent reading this manuscript.

In my opinion the novelty of the model suggested in this work is unquestionable. Contrary to previous works, it does not rest on hydraulic jump to explain the scours observed on the seafloor in CLTZs. I feel, however, that the authors should accentuate that previous hypotheses for the formation of scours in CLTZs might still be valid. That open recognition does not diminishes the power of the interpretation of this work; opposite to that, it makes the work more unique.

Below I included some comments to the authors, all minor. I expect the scientific community to receive this contribution with revealing curiosity.

Yours sincerely,

Maria Azpiroz-Zabala

General comments

-The authors present as their main conclusion a new mechanism that explain scour field formation at the termination of confining submarine channels. They acknowledge that other models, all based on hydraulic jump presence, have been suggested to describe the scour field formation. The authors concede that hydraulic jumps could be linked to scour field formation in line 179-180 '...the association of scour fields in CLTZs with hydraulic jumps remains open for debate.'. However, I felt when reading this work, that the authors were undermining the previously suggested models in contrast to their model (lines 160-179). Here, they often used hypothetical words when writing on interpretations different to theirs (examples 1 to 3) that contrast with the words used for their own model (examples 4 and 5).

1.Line 161 - '...scours would form...';

2.Lines 167-168 - '...a single hydraulic jump would form a single scour rather than ';

3.Line 170 - '...Dorrell et al. have evoke the presence...';

4.Lines 153-155 - '...increase of the near-bed velocity gradient (...) resulting in scour development.';

5.Line 181 'Flow relaxation is a mechanism that well explains the formation of scour fields...'.

I suggest that the authors make clearer to the reader that their model is a novel additional mechanism that explains scour field formation on the top of existing models.

-Following the previous comment, they authors cite bibliography on sediment-laden density flows showing hydraulic jumps that could potentially form scours. However, the authors specifically cite saline flows when referring to previous observations of hydraulic jumps at the termination of channels. As a reader, that seems to indicate that sediment-laden flows have not shown hydraulic jumps at the channel termination. This thought is reinforced in lines 173-174 '...it remains questionable whether such hydraulic-jump-array model translate across to turbidity currents.'. I agree that processes observed in saline flows might not occur in turbidity currents. However, to make clearer that such hydraulic jumps have been observed at the termination of channels confining turbidity currents, I think the authors should refer to these works (*), and rephrase lines 160-174 to show their exceptional findings and at the same time credit previous hypotheses.

(*) If the authors find it useful, I suggest Baas et al., 2004, Deposits of depletive high-density

turbidity currents: a flume analogue of bed geometry, structure and texture, *Sedimentology* 51, 1053-1088.

-The authors base their model on experiments in which the sudden loss of full confinement of the flow is key. They show bathymetric images where this sudden loss happens in the field (Figure 1e, 2006 survey). However, I would tend to think this is not ubiquitous and that confining walls would decrease their height when getting closer to the channel termination. In that case flows would overflow before getting completely unconfined and relaxation would be less extreme than in the cases in this work. If the authors agree with this idea, can they stress the importance that the sudden aspect of the loss of confinement has in flow deformation?

Additional comments

- Line 37 - '...which focus the flow and prevent deposition...'. I agree with the authors on that deposition is not the main process in channels although it can locally happen. Could the authors rephrase the sentence to suggest that deposition might still be possible in channels?

- Line 124-125 - '(d), Bed-elevation change during the experiments...'. Should the caption say 'Laterally averaged bed-elevation of a 30 cm wide strip...' similarly to caption in (c)?

- Line 169 - '...have revealed', should it be '...has revealed'?

- Line 184 - '...triggers scour formation.'. The authors seem to state that flow relaxation always triggers scour formation. I think this might not be the case in certain conditions of flow concentrations, velocities, compositions in combination with channel morphology. Could the authors rephrase their sentence to indicate that scours 'might be triggered'?

- Line 201 - '...maintains the flow thickness...'. Self-confinement would maintain the flow thickness if the flow is completely confined within the confinement walls. I would expect this to happen progressively. At the first stages, flows would start to initiate the propagation of the channel with overflowing of the flow until in a later stage the propagating channel would be incised deeper, and eventually fully confined flows would maintain their thickness. If the authors agree with this, could they rephrase the sentence? The flow thickness would be maintained at the final stage only, or alternatively, in the case of flows thinner than channel walls.

- Line 308-309 - '...this calculation assumes that the bed-normal component of velocity is zero.'. Can the authors assess the importance that this assumption has in their model? In their model, flow gets thinner and therefore velocities normal to the bed might be of relative importance?

- Line 315 - 'Time-averaged profiles...' should say 'Time-averaged velocity profiles...'.

- Line 315 - '...were generated for the body of the current,...'. The flow body has not been defined in this work, and in my opinion is out of its scope. I suggest the authors use here the actual range of time (~25-90 sg) used to average the velocity instead of any reference to the body of the flow.

- Line 316 - 'The velocity measurement of the current head and of the tail...'. See previous comment.

Figure 3a and b

-The figure shows a depositional area in the centre of the channel (from -1 m to 0.25 m in downstream distance) in both panels. This depositional area splits the base of the flow into two basal flows that converge approximately at the location of UVP 3. In panel a, UVP 3 is located almost coincident with the loss of flow confinement. Could the authors explain, maybe in the Materials and Methods section if the convergence of the two basal flows could produce a change in the downstream flow behaviour just at the loss of confinement and, therefore, affect relaxation?

-What the dark grey colour in panels a and b represent? If this represents no change of bed elevation I suggest that the dark grey colour is added to the colorbar of the panels.

Figure 3c - The panel shows the laterally averaged incision-depth of a 30 cm wide strip on the bed. Could the authors include in the Materials and Methods section a cross-section of both loss of confinement and reference experiments where the 30 cm strip is shown as representative for the interpretation?

Figure 3c and d, and Figure 4e - The Fig.3c and d show that the largest relative erosion of the loss of confinement experiment to the reference experiment occurs at UVP 5. The authors correlate erosion with an enhanced flow shear velocity due to flow relaxation. However, the ratio between shear velocity of loss of confinement and reference experiments is smaller in UVP 5 than, for example, UVP 4 (Fig. 4e). This might be because the values shown are time-/depth-/width-averages of the variables. It might be interesting to show, maybe in the Materials and Methods section, the evolution of the shear velocity at each UVP location. This might clarify the differences shown in Fig. 3 and Fig.4e and highlight the more erosive parts of the flow at each location.

Figure 4 - I suggest the authors refer in the caption to the Materials and Methods section for details on average calculation.

Figure S1 - Legend says 'Monetrey Canyon...', it should say 'Monterey Canyon...'.

Legend refers to Azpiroz-Zabala et al., 2017 work and states that the values shown refer to the body of the current in that work. I think figure S1 shows the expected values for the flows in the body of the Congo Canyon publication with same sediment size as the one used in this work. If this is right, the reference to the sediment size should be added to the figure/legend/caption.

Figure S4 - I suggest swapping the colour of the lines to keep consistency with figures 3 and 4.

Figure S5 - Equation in figure should say $u=U_{uvp}/\cos(\alpha)$.

Figure S6 - A qualitative observation of the figure seems to show that flow at UVP 4 is in average thinner than at UVP 5. However, the value in Figure 4d shows the opposite. It might be useful to have a supplementary figure with the dispersion of the thickness of the flow over the averaging period.

Reviewer #2 (Remarks to the Author):

This paper presents a new concept (named flow relaxation) to explain what happens when the confinement of submarine channels is lost at the channel to lobe transitions (CTLT). Flow relaxation is attributed to the development of lateral pressure gradients when there are no longer channel walls. They then support these ideas with tank test experiments, which are consistent with their flow relaxation model in explaining patterns of erosion and deposition. To me this is a very interesting idea and can see that this might become a seminal concept.

CTLT are presently a hot topic within the geological literature. The paper points out that the existing literature on what happens to generate scours within the CTLT is usually attributed to the occurrence of hydraulic jumps. They point out that there is no substantial confirmation that this is correct and that the main body of the literature is at risk of being inconsistent with some of the predictions associated with the hydraulic jump models. Although I am involved in some of the literature which piles-on the hydraulic jump interpretations, I am in complete agreement that something is wrong or missing in the application of these concepts to the limited available data. They are offering an appealing substitute concept to consider.

Comments and suggestions made while reading the ms:

Line 19 - "exactly" seems excessive.

Line 21 – Found meaning of "commonly linked" confusing. Suggest replacing it with "commonly hypothesized" and the "hypothesis" to "assertion" in next sentence.

Lines 35-36 – Implies plastic is preferentially concentrated in turbidite deposits. Question whether the connection between marine plastic debris and turbidity currents was actually made in either reference 4 or 5 (or elsewhere). Suggest cutting "also as a depot for plastic debris".

Line 50 - Might want to include or replace with Carvajal, C., et al. 2017, Unraveling the channel-to-lobe-transition zone with high-resolution AUV bathymetry: Navy Fan, offshore Baja California, *Journal of Sedimentary Research*, v. 87, p. 1049-1059, doi.org/10.2110/jsr.2017.58 as it includes much higher resolution of what a CTLT actually looks like rather than several of the older references.

Page 5 figure 1. This figure should be improved. Part of this is that just too much is crammed into the space. A cartoon something like part a is probably needed, however seems obviously grabbed from elsewhere as it included several features not explicitly discussed in this paper and not specifically designed to capture the issues addressed in this paper. Not sure why the global map shown in part b is needed at all? Parts c and d are at regional scales, hard to see the scours and there is no indication of channel propagation. Moreover, the morphology shown on the margin associated with Penghu Canyon (miss-drafted as Cannyon) seems to have scour like features throughout the channel and elsewhere on the margin. Part e seems to support the case for the propagating channel, but in style is a poor match with the existing cartoon in part a. Cut "Northern" and "Central" and probably unexplained dotted lines. Recommend enlarging and including just essence of a and e. Also, the textures shown in part e 2006 and 2011 surveys are quite different. Is this because the later survey used a higher frequency multibeam sonar?

Line 56 – On initial reading whether "The favored hypothesis" is referring to the explanations in the literature or in this paper. Suggest clarifying.

Line 64 and or Figure 2. Suggest someplace explicitly saying this is a "test tank" or "flume" experimental set up.

Figure 3 In parts a and b the ovals of no bed elevation changes surrounding UVP 1 and 2 have this charcoal colored area separating deposition and erosion, which is not included in the key.

Lines 181-187. The test tank result presented here does seem to explain erosion at the CTLT. However, in the following sentences, it seems to be going on from the formation of individual scours to scour fields rather quickly, passing it off simply as being related to "irregularities and homogeneities on the ocean floor". Suggest framing this jump better. Why is there roughness here in the first place? Moreover, is Allen 1971 a relevant reference for this?

Line 188 – The paragraph indentation/separation was lost in the compiled pdf. However, I am assuming this is the beginning of at least a new paragraph. However, might be worth a new subheading as the text shifts gear to channel growth.

Lines 191, 206, -Three different phrases for the Squamish (system, Delta, Prodelta).

Lines 133-135 and 136 to 137. Wording describing both c and e are unclear.

Paragraph in lines 160-180. This seems to be somewhat redundant with material introduced earlier (i.e., abstract and lines 56-62). As is, it is focused on the previous models, but does not integrate back to the new concept presented in the text. Suggest either moving to introduction.

Line 206 Suggest changing to "A bathymetric survey of the Squamish Delta, that was conducted in 2006," ...

Line 208 Suggest changing "study" to "survey".

Line 210 – It is rather emphatically stated that channel grow at Squamish was driven by erosion. While I believe it is likely, would be nice to have a reference from the Squamish literature or a difference map that further supports this interpretation?

Line 230- 287 – While the Shields scaling is billed in the Abstract Lines 22-24, and Introduction line 62 as being new and presumably important, it was not mentioned again until this line in the Methods section. While I think the purpose of these calculations is to indicate that the design of the test tank experiment was a priori calculated to be appropriate. However, in the end, there was deposition and erosion occurring in the test tank experiment. Thus, the impact of this to the results of the experiment and ultimately on the conclusions of the manuscript was not explained. I wonder why it is included at all?

Reviewer #3 (Remarks to the Author):

Review – Pohl et al. "New flow relaxation mechanism explains scour fields at the end of submarine channels"

The finding of this new experiment will have wide interest in the deep-sea sedimentary community. Unlike fluvial systems, our ability to observe field-scale turbidity current flow evolution over time and space is strongly limited by logistics (deep ocean, low periodicity, massively destructive) and thus appropriately scaled tank experiments, such as this, are a huge step forward..

These findings are an important result that follows on from a preceding nature-comm paper ~ 3 years ago (deLeeuw et al.). The group at Delft appear to have succeeded in managing the Shields scaling requirement problems that have limited previous tank experiments (a balance of slope, SSC and suitable grain roughness apparently –well explained in methods). That previous paper showed the autogenic nature of progressive confinement within channels through both differential accretion and erosion. This paper extends that exciting result to describe the, again autogenic means by which flows can erode just beyond the channel mouths thereby prograding the channel system.

Taken together, these two papers for first time address the previously unconstrained speculation that has accompanied descriptions of observed relict deep-sea morphologies. Included in this speculation is a recent spate of papers that have built on each other citing the (still unproven) importance of assumed hydraulic jumps at these channel lobe transition zones (CLTZ). This new work, in contrast, demonstrates that there is no need for this inference and that a surprisingly non-intuitive, and previously unsuspected, mechanism can explain much of the observed morphology.

An important aspect is that these tank-scale experiments allow one to view the basal boundary layer which has never adequately been measured in the few field-scale current profiles. This paper presents clear evidence of a drop in the velocity maximum with a corresponding increase in the local bed shear stress. Having multiple closely spaced UVPS through the CLTZ, the onset and temporal and spatial evolution of this critical period of erosion is seen for the first time.

In the discussion they cite the lateral pressure gradients due to the loss of confinement as the cause of this flow "relaxation" (wouldn't be my favourite term for this new mechanism, but not inaccurate). As they don't have the UVPs spaced laterally, they don't see it in this instance, but they refer to the 2016 paper again which does have exactly that geometry.

This contribution is clearly described and I don't thus see a need to recommend changes in that. I have a few questions, concerning their discussion which they might be able to address (in the

limited # of words allowed):

- They don't mention measured suspended sediment concentration (SSC) in the resulting flow at any point – I don't think the UVP scattering can provide this. In the previous paper they merely assumed that it didn't change from the incoming suspension (17% in this case). Do they assume the same or suspect a loss or gain in SSC?
- Does the UVP scattering provide any indication as to whether the vertical SSC profile similarly collapse with the velocity maximum? Fig 4d shows the flow "thinning" but the plotted thickness of the flow is based on where the velocity is $\frac{1}{2}$ that of the peak – not the thickness of the suspended sediment layer – could the UVP confirm that the height to clear water also shrank?
- They are careful not to describe this flow as either super or sub-critical. I think this is wise given the popular bandwagon of use/misuse of these terms. They do state however that, by virtue of there being no hydraulic jump, the flow state doesn't change – would they be confident to say that it remain supercritical throughout?
- They mention a topographic "noise" in the incision depth that was in their laser-scanned post-flow drained topography. Would they comment on whether this roughness had any periodicity that might be indicative of scour spacing or bedforms? Would such larger roughness elements even have any scaleable meaning?
- This particular flow exhibited the all-important short distance downstream of the loss of confinement where the shear velocity of the unconfined flow was greater than the confined flow (Fig 4 e). This occurs even though the flow is already significantly slower than the confined one (Fig 4c). As far as I can see this experiment was only run once (each for confined and unconfined). (Is this true?) Would they speculate whether, with a slight change in either the input slope or SSC (but still meeting their Shield scaling) that this zone of enhanced erosion would get longer or shorter?
- While I think the concept is clearly demonstrated with this (singular?) tank experiment, it is just a result of a single instance of the flow. The flow matched the preexisting topography in the sense that it was completely constrained by the "levees" upstream and had sufficient momentum to not decelerate too rapidly. Any real CLTZ will, however, receive a wide range of flow competences. Some will exhibit this mechanism, others may die at this transition, and yet others be already out of equilibrium with the preexisting channel (already overwhelming the relief). Would they be prepared to comment on how varying incoming flow competencies might alter the presence and/or length scale over which this relaxation extends?

These questions are perhaps more for my curiosity. I do not insist that they be answered. I believe the paper stands already as acceptable by Nature Comms. I look forward to it coming out.

John HC

The line numbers given in these replies refer to the file 'MS-Tracked.pdf'.

Reviewer #1 (Remarks to the Author):

The paper by Pohl and co-authors presents an original mechanism, termed relaxation by the authors, to explain scour field formation at channel-lobe transition zones (CLTZs). Previous works have suggested scour formation at the termination of submarine channels as a result of processes linked to hydraulic jumps. This is an exciting contributing that, in contrast to those previous hypotheses, identifies a new process able to form the scour fields. I would like to congratulate the authors on their clear explanation and supporting figures, and thank them for the really satisfying time I spent reading this manuscript. In my opinion the novelty of the model suggested in this work is unquestionable. Contrary to previous works, it does not rest on hydraulic jump to explain the scours observed on the seafloor in CLTZs. I feel, however, that the authors should accentuate that previous hypotheses for the formation of scours in CLTZs might still be valid. That open recognition does not diminishes the power of the interpretation of this work; opposite to that, it makes the work more unique.

Below I included some comments to the authors, all minor. I expect the scientific community to receive this contribution with revealing curiosity.

Yours sincerely,

Maria Azpiroz-Zabala

We thank the reviewer for her kind words and for the time she spend on reviewing our manuscript. We are confident that the modification made here in response to her comments will clarify and improve the quality of our work.

We have modified the text to further accentuate the previous hypotheses as requested above by the reviewer (see detailed explanation below).

General comments

-The authors present as their main conclusion a new mechanism that explain scour field formation at the termination of confining submarine channels. They acknowledge that other models, all based on hydraulic jump presence, have been suggested to describe the scour field formation. The authors concede that hydraulic jumps could be linked to scour field formation in line 179-180 '...the association of scour fields in CLTZs with hydraulic jumps remains open for debate.'. However, I felt when reading this work, that the authors were undermining the previously suggested models in contrast to their model (lines 160-179). Here, they often used hypothetical words when writing on interpretations different to theirs (examples 1 to 3) that contrast with the words used for their own model (examples 4 and 5).

- 1.Line 161 - '...scours would form...';
- 2.Lines 167-168 - '...a single hydraulic jump would form a single scour rather than ';
- 3.Line 170 - '...Dorrell et al. have evoke the presence...';
- 4.Lines 153-155 - '...increase of the near-bed velocity gradient (...) resulting in scour development.';
- 5.Line 181 'Flow relaxation is a mechanism that well explains the formation of scour fields...'

I suggest that the authors make clearer to the reader that their model is a novel additional mechanism that explains scour field formation on the top of existing models.

We thank the reviewer for raising this issue. It is not our intention to dismiss previous models explaining the formation of scour fields in CLTZs. We follow the suggestions by the reviewer and use a more objective phrasing for the explanation of previous models and our own model.

Changes applied to the manuscript:

1. Line 178-179:

~~“In this model scours would form because of enhanced turbulence and created by a hydraulic jump.”~~

Sentence replaced with:

“...which cause enhanced turbulence and consequently, increased erosion”

2. Line 191-192:

~~“... a single hydraulic jump would forms a single scour rather than...”~~

3. Line 194-195:

~~“...Dorrell et al. ¹⁴ have proposed evoked the presence...”~~

4. Line 170-171:

~~“...increase of the near-bed velocity gradient (...) triggering resulting in scour development.”~~

5. Line 206-207:

~~“Flow relaxation provides a novel is a mechanism to ~~that well~~ explains the formation of scour fields in CLTZs.”~~

-Following the previous comment, they authors cite bibliography on sediment-laden density flows showing hydraulic jumps that could potentially form scours. However, the authors specifically cite saline flows when referring to previous observations of hydraulic jumps at the termination of channels. As a reader, that seems to indicate that sediment-laden flows have not shown hydraulic jumps at the channel termination. This thought is reinforced in lines 173-174 '...it remains questionable whether such hydraulic-jump-array model translate across to turbidity currents.'. I agree that processes observed in saline flows might not occur in turbidity currents. However, to make clearer that such hydraulic jumps have been observed at the termination of channels confining turbidity currents, I think the authors should refer to these works (*), and rephrase lines 160-174 to show their exceptional findings and at the same time credit previous hypotheses.

(*) If the authors find it useful, I suggest Baas et al., 2004, Deposits of depletive high-density turbidity currents: a flume analogue of bed geometry, structure and texture, *Sedimentology* 51, 1053-1088.

As raised in the previous comment of the reviewer, we have rephrased parts of the sentences in line 160-174 (original manuscript) to describe the different models more objectively. See previous comment.

We now acknowledge the experiments of Baas et al. (2004), and we also include a description of these experiments in the manuscript. However, the occurrence of the hydraulic jump in these experiments cannot be associated with the loss of confinement because the channel termination was accompanied with a slope break (a decrease of the flume tank floor). A slope break is likely to trigger the formation of a hydraulic jump as demonstrated by previous experiments on turbidity current crossing a slope break (e.g. Garcia & Parker, 1989; Garcia 1993; Islam & Imran 2010).

Therefore, it is not clear whether the hydraulic jump in the experiments by Baas et al. (2004) was triggered by a slope break or a loss of confinement. In our experiments on the other hand, the turbidity current solely flows through a loss of confinement without a slope break.

Changes applied to the manuscript:

Additional text: Line 186-191:

“Hydraulic jumps downstream of a loss of confinement have been observed in particle -laden density flows (turbidity currents) in experiments by Baas et al. ³². However, the loss of confinement in those experiments was accompanied with a sudden decrease in slope gradient. Hydraulic jumps are known to frequently occur due to sudden decreases of the slope ^{31,33,34}. Therefore, it seems more likely that the hydraulic jump in these experiments by Baas et al. ³² was triggered by the break in slope rather than the loss of confinement.”

References:

Garcia, M. & Parker, G. 1989. Experiments on hydraulic jumps in turbidity currents near a canyon fan transition. *Science*, 245, 393–396, <https://doi.org/10.1126/science.245.4916.393>.

Garcia, M.H. 1993. Hydraulic Jumps in Sediment-Driven Bottom Currents. *Journal of Hydraulic Engineering*, 119, 1094–1117, [https://doi.org/10.1061/\(ASCE\)0733-9429\(1993\)119:10\(1094\)](https://doi.org/10.1061/(ASCE)0733-9429(1993)119:10(1094)).

Islam, M.A. & Imran, J. 2010. Vertical structure of continuous release saline and turbidity currents. *Journal of Geophysical Research: Oceans*, 115, 1–14, <https://doi.org/10.1029/2009JC005365>.

-The authors base their model on experiments in which the sudden loss of full confinement of the flow is key. They show bathymetric images where this sudden loss happens in the field (Figure 1e, 2006 survey). However, I would tend to think this is not ubiquitous and that confining walls would decrease their height when getting closer to the channel termination. In that case flows would overspill before getting completely unconfined and relaxation would be less extreme than in the cases in this work. If the authors agree with this idea, can they stress the importance that the sudden aspect of the loss of confinement has in flow deformation?

We thank the reviewer for this comment. Our experiments show that flow relaxation persists when a more gradual loss of confinement is established. We now specifically mention this gradual transition and its consequences for natural systems.

Changes applied to the manuscript:

Added text, line 240-247:

“Channel propagation in the experiment resulted in a more gradual loss of confinement, which still triggered flow relaxation and erosion. Additionally, channel propagation in natural systems will depend on the size of the turbidity currents with respect to the channel dimensions. If turbidity currents are too small to reach the end of the channel, flow relaxation cannot occur and the channel will not propagate, as seen for the Southern Channel in the Squamish Delta between 2004 and 2006 ^{26,44}. On the other hand, if a turbidity current is too large for the channel, then the flow will overspill the channel, thereby reducing

its size until the flow matches the channel dimensions⁴⁵, in which case flow relaxation will occur.”

Additional comments

- Line 37 - '...which focus the flow and prevent deposition...'. I agree with the authors on that deposition is not the main process in channels although it can locally happen. Could the authors rephrase the sentence to suggest that deposition might still be possible in channels?

We agree with the reviewer that deposition within channels is still be possible. To make this clearer we have rephrased the sentence in line 37 (original manuscript).

Changes applied to the manuscript:

Line 42-43:

“...which focus the flow and predominately prevent deposition...”

- Line 124-125 - '(d), Bed-elevation change during the experiments...'. Should the caption say 'Laterally averaged bed-elevation of a 30 cm wide strip...' similarly to caption in (c)?

The reviewer’s comment indicates that the figure caption was not clear enough. We now specify the spatial extent of the measurement in the caption.

Changes applied to the manuscript:

Line 136-138:

“(d) Bed-elevation change during the experiments as captured in the footprint of the UVP-probes (see Materials and Methods) ~~by the UVPs~~.”

- Line 169 - '...have revealed', should it be '...has revealed'?

We corrected this typo.

- Line 184 - '...triggers scour formation.'. The authors seem to state that flow relaxation always triggers scour formation. I think this might not be the case in certain conditions of flow concentrations, velocities, compositions in combination with channel morphology. Could the authors rephrase their sentence to indicate that scours 'might be triggered'?

We agree with the reviewer and now state that the potential of scour formation is increased and no longer imply the necessary formation of scours.

Changes applied to the manuscript:

Line 209-211:

“This increase of the erosional potential of the sediment bed by the flow also and increases the potential for triggers scour formation over the entire area in which the flow relaxes³⁴.”

- Line 201 - '...maintains the flow thickness...'. Self-confinement would maintain the flow thickness if the flow is completely confined within the confinement walls. I would expect this to happen progressively. At the first stages, flows would start to initiate the propagation of the channel with overspilling of the flow until in a later stage the propagating channel would be incised deeper, and eventually fully confined flows would maintain their thickness. If the authors agree with this, could they rephrase the sentence? The flow thickness would be maintained at the final stage only, or alternatively, in the case of flows thinner than channel walls.

We agree with the reviewer that establishing self-confinement is a gradual process that leads to a progressive decrease of flow spreading. We have modified the text to highlight the progressive nature of this process.

Changes applied to the manuscript:

Line 234-236:

"The increasing levels of self-confinement start to provide lateral support to the flow, which results in a decrease of the lateral pressure gradient, and prevent spreading of the lower part of the flow maintains the flow thickness."

- Line 308-309 - '...this calculation assumes that the bed-normal component of velocity is zero.'. Can the authors assess the importance that this assumption has in their model? In their model, flow gets thinner and therefore velocities normal to the bed might be of relative importance?

The reviewer is right that the bed normal velocities might play a role when the flow is thinning or thickening. However, the average bed normal velocity component of the flow can be neglected. Based on the decrease in the flow thickness over time, the average bed normal velocity would be ~ 1 cm/s and is, therefore, significantly lower than the bed parallel velocity which is with > 50 cm/s. We have now added this information to the Materials and methods sections for clarification.

Changes applied to the manuscript:

Additional text: Line 364-367:

"Thickness changes of the turbidity current over time suggest a bed-normal velocity component of ~ 0.01 m s⁻¹. This value is 50 times smaller than the minimum depth-averaged velocity of 0.5 m s⁻¹ and validates the neglect of the bed-normal velocity component."

- Line 315 - 'Time-averaged profiles...' should say 'Time-averaged velocity profiles...'.

Thank you for pointing this out. We rephrased the sentence accordingly.

- Line 315 - '...were generated for the body of the current,...'. The flow body has not been defined in this work, and in my opinion is out of its scope. I suggest the authors use here the actual range of time (~ 25 -90 sg) used to average the velocity instead of any reference to the body of the flow.

We agree with the reviewer that the definition of the current head, body, and tail are not the scope of this paper. Therefore, we removed these terms from the manuscript and mention the specific time interval as suggested by the reviewer.

Changes applied to the manuscript:

Line 373-377:

“Time-averaged velocity profiles were generated over a time interval for the body of the current, where the flow is was generally steady (Fig. 4a and b). The time-averaging interval started 10 seconds after the arrival of the turbidity current at the measurement location and was continued over 65 seconds. The velocity measurement of the current head and of the tail were omitted for the time-averaging (Supplementary Figures 6 and 7).”

- Line 316 - 'The velocity measurement of the current head and of the tail...'. See previous comment.

See previous comment.

Figure 3a and b

-The figure shows a depositional area in the centre of the channel (from -1 m to 0.25 m in downstream distance) in both panels. This depositional area splits the base of the flow into two basal flows that converge approximately at the location of UVP 3. In panel a, UVP 3 is located almost coincident with the loss of flow confinement. Could the authors explain, maybe in the Materials and Methods section if the convergence of the two basal flows could produce a change in the downstream flow behaviour just at the loss of confinement and, therefore, affect relaxation?

We thank the reviewer for this comment. The experiments show that the deposition in the proximal part of the channel (from -1 to 0.25m) has no impact on the flow relaxation mechanism. As the depositional area occurred in both the loss of confinement and the reference experiment (Fig. 3a). Thus, the increase in erosion observed in the loss of confinement experiment cannot be associated with the deposition in the proximal part of the channel.

-What the dark grey colour in panels a and b represent? If this represents no change of bed elevation I suggest that the dark grey colour is added to the colorbar of the panels.

We thank the reviewer for indicating the missing color in the colourbar. We have now added the grey color to the colourbar in figure 3. This comment was also raised by reviewer 2. The dark grey color indicates a change in bed elevation below 3 mm.

Figure 3c - The panel shows the laterally averaged incision-depth of a 30 cm wide strip on the bed. Could the authors include in the Materials and Methods section a cross-section of both loss of confinement and reference experiments where the 30 cm strip is shown as representative for the interpretation?

We agree with the reviewer that the location of the 30 cm strip was unclear. We have now indicated the 30 cm strip in figure 3.

Figure 3c and d, and Figure 4e - The Fig.3c and d show that the largest relative erosion of the loss of confinement experiment to the reference experiment occurs at UVP 5. The authors correlate erosion with an enhanced flow shear velocity due to flow relaxation. However, the ratio between shear velocity of loss of confinement and reference experiments is smaller in UVP 5 than, for example, UVP 4 (Fig. 4e). This might be because the values shown are time-/depth-/width-averages of the variables. It might be interesting to show, maybe in the Materials and Methods section, the evolution of the shear velocity at each UVP location. This might clarify the differences shown in Fig. 3 and Fig.4e and highlight the more erosive parts of the flow at each location.

We agree with the reviewer the evolution of the shear velocity values over time would be of great value. Unfortunately, our measurements do not allow us to generate these plots. Instantaneous velocity profiles are noisy and therefore unsuitable to calculate shear velocity values. Time averaging is required to calculate smooth velocity profiles suitable to calculate shear velocity values. In our experiments this time averaging window needs to be longer than 40 seconds, which prevents us from showing the shear velocity evolution through time.

Figure 4 - I suggest the authors refer in the caption to the Materials and Methods section for details on average calculation.

We follow the reviewer's advice and refer now to the Materials and Methods.

Figure S1 - Legend says 'Monetrey Canyon...', it should say 'Monterey Canyon...'.
The reviewer is right. For the calculation values for the Congo Canyon turbidity current in the Shields mobility diagram we used a grain size of 200 μm , which is in the grain size range of the sediment we used in the experiments. We added this information in the manuscript.

We corrected this typo.

Legend refers to Azpiroz-Zabala et al., 2017 work and states that the values shown refer to the body of the current in that work. I think figure S1 shows the expected values for the flows in the body of the Congo Canyon publication with same sediment size as the one used in this work. If this is right, the reference to the sediment size should be added to the figure/legend/caption.

The reviewer is right. For the calculation values for the Congo Canyon turbidity current in the Shields mobility diagram we used a grain size of 200 μm , which is in the grain size range of the sediment we used in the experiments. We added this information in the manuscript.

Changes applied to the manuscript:

Line 399-400:

"For calculation of ~~the point for~~ the Congo Canyon, the body of the current and a grain size of 200 μm was used"

Figure S4 - I suggest swapping the colour of the lines to keep consistency with figures 3 and 4.

We thank the reviewer for pointing out this inconsistency in our initial colour choice. We have modified the colours accordingly.

Figure S5 - Equation in figure should say $u = U_{vp} / \cos(\alpha)$.

The reviewer is correct, and we thank her for pointing this out. We changed the equation accordingly.

Figure S6 - A qualitative observation of the figure seems to show that flow at UVP 4 is in average thinner than at UVP 5. However, the value in Figure 4d shows the opposite. It might be useful to have a supplementary figure with the dispersion of the thickness of the flow over the averaging period.

We thank the reviewer for pointing this out. We have made a change in the data presentation in Supplementary Figures 6 and 7, which now display the bed-parallel velocity, rather than the velocity in the direction of the respective probes. This change has mitigated the apparent flow thickness discrepancy observed by the reviewer.

Additionally, we have double-checked the calculated flow thicknesses of the two probes mentioned by the reviewer (Figure 1 below). This check demonstrated that the flow thickness does indeed decrease between UVP 4 and 5 in the experiment with loss of confinement, as presented in the original figure 4d of our manuscript.

Figure 1: Time-averaged velocity profiles of UVP 4 and 5. The velocity profiles measured by both UVP-probes are very similar. The time-averaged profiles were smoothed using a Fourier fitting function. The asterisk marks the velocity maximum and the triangle the calculated flow thickness.

Reviewer #2 (Remarks to the Author):

This paper presents a new concept (named flow relaxation) to explain what happens when the confinement of submarine channels is lost at the channel to lobe transitions (CTLT). Flow relaxation is attributed to the development of lateral pressure gradients when there are no longer channel walls. They then support these ideas with tank test experiments, which are consistent with their flow relaxation model in explaining patterns of erosion and deposition. To me this is a very interesting idea and can see that this might become a seminal concept.

CTLT are presently a hot topic within the geological literature. The paper points out that the existing literature on what happens to generate scours within the CTLT is usually attributed to the occurrence of hydraulic jumps. They point out that there is no substantial confirmation that this is correct and that the main body of the literature is at risk of being inconsistent with some of the predictions associated with the hydraulic jump models. Although I am involved in some of the literature which piles-on the hydraulic jump interpretations, I am in complete agreement that something is wrong or missing in the application of these concepts to the limited available data. They are offering an appealing substitute concept to consider.

We thank the reviewer for the constructive comments and appreciate the time spend on reviewing our work.

Comments and suggestions made while reading the ms:

Line 19 - "exactly" seems excessive.

We agree with the reviewer and removed the word 'exactly'.

Line 21 – Found meaning of "commonly linked" confusing. Suggest replacing it with "commonly hypothesized" and the "hypothesis" to "assertion" in next sentence.

We followed the reviewer's suggestion and rephrased the text accordingly.

Lines 35-36 – Implies plastic is preferentially concentrated in turbidite deposits. Question whether the connection between marine plastic debris and turbidity currents was actually made in either reference 4 or 5 (or elsewhere). Suggest cutting "also as a depot for plastic debris".

We initially cited the papers of Pham et al, (2014) and Cressey (2016) as they were addressing the abundance of plastic debris in marine sediments, in particular within canyons, which suggest transport and deposit of plastic debris by turbidity currents. This is supported by the recent paper from Kane & Clare (2019). We updated the reference and rephrased the above mention sentence accordingly.

Changes applied to the manuscript:

Line 41:

"..., and also as a potential depot for plastic debris 4."

References:

⁴ Kane, I. A. & Clare, M. A. 2019. Dispersion, accumulation and the ultimate fate of microplastics in deep-marine environments: A review and future directions. *Front. Earth Sci.* 7, 80.
<http://dx.doi.org/10.3389/feart.2019.00080>

Line 50 - Might want to include or replace with Carvajal, C., et al. 2017, Unraveling the channel-to-lobe-transition zone with high-resolution AUV bathymetry: Navy Fan, offshore Baja California, *Journal of Sedimentary Research*, v. 87, p. 1049-1059, doi.org/10.2110/jsr.2017.58 as it includes much higher resolution of what a CTLT actually looks like rather than several of the older references.

We would like to thank the reviewer for pointing out the relevance of the above-mentioned paper. We added the reference in line 56.

Page 5 figure 1. This figure should be improved. Part of this is that just too much is crammed into the space. A cartoon something like part a is probably needed, however seems obviously grabbed from elsewhere as it included several features not explicitly discussed in this paper and not specifically designed to capture the issues addressed in this paper. Not sure why the global map shown in part b is needed at all? Parts c and d are at regional scales, hard to see the scours and there is no indication of channel propagation. Moreover, the morphology shown on the margin associated with Penghu Canyon (miss-drafted as Cannyon) seems to have scour like features throughout the channel and elsewhere on the margin. Part e seems to support the case for the propagating channel, but in style is a poor match with the existing cartoon in part a. Cut “Northern” and “Central” and probably unexplained dotted lines. Recommend enlarging and including just essence of a and e. Also, the textures shown in part e 2006 and 2011 surveys are quite different. Is this because the later survey used a higher frequency multibeam sonar?

We thank the reviewer for the remarks on figure 1. We have redrafted this figure in response to the reviewer’s suggestions. Panels b, c, and d have been removed to improve the clarity and to make the figure less crammed. The other panels are enlarged to improve appearance and readability. We changed the references in the text following the comments of the reviewer.

The annotations ‘Northern’, ‘Central’, and the dotted lines were from the publication where we took the bathymetry map from. We now have a bathymetry map with a higher resolution.

Line 56 – On initial reading whether “The favored hypothesis” is referring to the explanations in the literature or in this paper. Suggest clarifying.

Thanks for pointing out this unclarity. We rephrased the sentence.

Changes applied to the manuscript:

Line 62:

“The favored hypothesis within the literature to explaining erosion...”

Line 64 and or Figure 2. Suggest someplace explicitly saying this is a “test tank” or “flume” experimental set up.

We agree with the reviewer that this should be more explicit in the text.

Changes applied to the manuscript:

Line 72:

“...lateral confinement in a flume experiment-set-up (Fig. 2a).”

Figure 2 caption; Line 91-92:

“The flume experiment with loss of confinement.”

Figure 2 caption; Line 93:

“The reference flume experiment with”

Figure 3 In parts a and b the ovals of no bed elevation changes surrounding UVP 1 and 2 have this charcoal colored area separating deposition and erosion, which is not included in the key.

We thank the reviewer for pointing this out. The same point was raised by reviewer 1. The dark grey color indicates a change in bed elevation below 3 mm, which we now added to the colourbar.

Lines 181-187. The test tank result presented here does seem to explain erosion at the CTLT. However, in the following sentences, it seems to be going on from the formation of individual scours to scour fields rather quickly, passing it off simply as being related to “irregularities and homogeneities on the ocean floor”. Suggest framing this jump better.

We thank the reviewer for highlighting this unclarity in the text. We rephrased the above-mentioned section and explain the association of flow relaxation with the formation of the scours and scour fields in more detail.

Changes applied to the manuscript:

Line 184-187 (original manuscript):

~~“Hence, the likelihood of the formation of scours is increased over the entire area in which the flow relaxes. In this area, the locations of individual scours are likely determined by irregularities and inhomogeneities on the ocean floor³⁴, thereby explaining the observed scour fields in CLTZs.~~

Replaced with:

Line 211-216:

“Experiments have shown that erosive conditions can trigger the formation of fields of individual scours, in which the location of individual scours is controlled by random irregularities on the ocean floor^{39,40}. Such irregularities could consist of patches of coarser grains, a series of small-scale bedforms, a scour surrounding a boulder, or any biological feature. The formation of individual scours at these irregularities would then result in the formation of scour fields.”

References:

- ³⁹ Coleman, J. C. An indicator of water-flow in caves. Univ. Bristol speleol. Soc. 6, 57–67 (1949).

Why is there roughness here in the first place?

We thank the reviewer for raising this question, and now give examples for possible causes of irregularities on the ocean floor. (see also applied changes as in previous comment)

Changes applied to the manuscript:

Additional text; Line 213-215:

“Such irregularities could consist of patches of coarser grains, a series of small-scale bedforms, a scour surrounding a boulder, or any biological feature”

Moreover, is Allen 1971 a relevant reference for this?

We now also refer to the original paper by Coleman (1949) proposing the formation of scours at irregularities. (applied changes as in previous comment)

Line 188 – The paragraph indentation/separation was lost in the compiled pdf. However, I am assuming this is the beginning of at least a new paragraph. However, might be worth a new subheading as the text shifts gear to channel growth.

That is correct, line 188 (original manuscript) was the beginning of a new paragraph. We now use an indentation for the first line of each paragraph.

Lines 191, 206, -Three different phrases for the Squamish (system, Delta, Prodelta).

We thank the reviewer for pointing out this inconsistency in our terminology. We now use only the phrase “Squamish Delta” throughout the entire manuscript.

Changes applied to the manuscript:

Line 224:

“...in the much smaller Squamish ~~system~~ Delta.”

Line 248-249:

“...in the Squamish ~~Pre~~Delta.”

Lines 133-135 and 136 to 137. Wording describing both c and e are unclear.

We agree with the reviewer that the phrasing in the figure caption might be unclear and could be misunderstood. We rephrased the figure caption accordingly.

Changes applied to the manuscript:

Caption of figure 4, panel (c); Line 145-146:

“The turbidity current downstream of the loss of confinement ~~the turbidity current~~”

~~decelerated and was slower than the turbidity current in the reference experiment.~~

Caption of figure 4, panel (d); Line 147-148:

~~“After leaving the confinement the calculated turbidity current thickness decreased immediately thinned.”~~

Caption of figure 4, panel (e); Line 149-151:

~~“Shear velocity was slightly increased downstream of the loss of confinement was increased in comparison with the reference experiment and decreased farther downstream.”~~

Paragraph in lines 160-180. This seems to be somewhat redundant with material introduced earlier (i.e., abstract and lines 56-62). As is, it is focused on the previous models, but does not integrate back to the new concept presented in the text. Suggest either moving to introduction.

We understand the comment of the reviewer and reduced the redundancy of the information given in the introduction and discussion.

In the introduction we introduce the hydraulic jump as a commonly used mechanism explaining the formation of scours. This mechanism is discussed in more detail in the discussion and conclusion section.

Changes applied to the manuscript:

We removed the sentence in line 60-61 (original manuscript):

~~“Russell and Arnott²⁴ explain scouring in the CLTZ by the impingement of vortices that were produced by the hydraulic jump.”~~

Line 206 Suggest changing to “A bathymetric survey of the Squamish Delta, that was conducted in 2006,” ...

We agree with the reviewer and rephrased the text accordingly.

Changes applied to the manuscript:

Line 249:

~~“A bathymetric survey y map of the Squamish Delta, that was monitored conducted in 2006,”~~

Line 208 Suggest changing “study” to “survey”.

We rephrased the text accordingly.

Line 210 – It is rather emphatically stated that channel grow at Squamish was driven by erosion. While I believe it is likely, would be nice to have a reference from the Squamish literature or a difference map that further supports this interpretation?

We agree with the reviewer and now include a reference to the recent paper by Vendettuoli et al. (2019), which demonstrates the occurrence of erosion downstream of the loss of confinement of the Southern Channel on Squamish Delta.

Changes applied to the manuscript:

Line 252-256:

“Channel propagation was generated by the incision into the underlying substrate downstream of the rapid loss of confinement as demonstrated by 93 daily repeat surveys performed in 2011 ⁴⁴. ~~And, hence, propagation of the Southern channel was driven by erosion comparable to the channel propagation in our experiments (Fig. 3a).~~”

References:

- ⁴⁴ Vendettuoli, D., Clare, M. A., Clarke, J. H., Vellinga, A., Hizzet, J., Hage, S., ... & Stacey, C. (2019). Daily bathymetric surveys document how stratigraphy is built and its extreme incompleteness in submarine channels. *Earth and Planetary Science Letters*, 515, 231-247. <https://doi.org/10.1016/j.epsl.2019.03.033>

Line 230- 287 – While the Shields scaling is billed in the Abstract Lines 22-24, and Introduction line 62 as being new and presumably important, it was not mentioned again until this line in the Methods section. While I think the purpose of these calculations is to indicate that the design of the test tank experiment was a priori calculated to be appropriate. However, in the end, there was deposition and erosion occurring in the test tank experiment. Thus, the impact of this to the results of the experiment and ultimately on the conclusions of the manuscript was not explained. I wonder why it is included at all?

We thank the reviewer for this constructive comment and we have modified the text to clarify the importance and nature of the Shield scaling approach.

Shields scaling allows flows in flumes to not only deposit but also erode sediment. It is indeed a common misconception that Shields scaling only leads to erosion. Shields scaling allows the flow to alternate between erosion and deposition in space and time, thereby enabling reproduction of natural seascapes more realistically. To prevent misunderstanding, we now explicitly state that Shields scaling allows for both deposition and erosion.

The development of the Shields scaling was published by de Leeuw et al (2016) and is not the main focus of the present paper. Therefore, a more detailed explanation of the Shields scaling is only provided in the Materials and Methods section

Changes applied to the manuscript:

Line 70-71:

“Here we use the newly developed Shields scaling approach ²⁵ to directly observe the pattern of erosion and deposition resulting from a turbidity current...”

Line 153-155:

“The Shields scaling approach allows us to link ~~As previously noted,~~ the morphological changes at the loss of confinement ~~result into~~ rapid flow deformation, and its associated ~~which in turn triggers~~ enhanced erosion.”

References:

- de Leeuw, J., Eggenhuisen, J. T. & Cartigny, M. J. B. Morphodynamics of submarine channel inception revealed by new experimental approach. *Nat. Commun.* 7, 1–7 (2016). <https://doi.org/10.1038/ncomms10886>

Reviewer #3 (Remarks to the Author):

Review – Pohl et al. “New flow relaxation mechanism explains scour fields at the end of submarine channels”

The finding of this new experiment will have wide interest in the deep-sea sedimentary community. Unlike fluvial systems, our ability to observe field-scale turbidity current flow evolution over time and space is strongly limited by logistics (deep ocean, low periodicity, massively destructive) and thus appropriately scaled tank experiments, such as this, are a huge step forward.

These findings are an important result that follows on from a preceding nature-comm paper ~ 3 years ago (deLeeuw et al.). The group at Delft appear to have succeeded in managing the Shields scaling requirement problems that have limited previous tank experiments (a balance of slope, SSC and suitable grain roughness apparently –well explained in methods). That previous paper showed the autogenic nature of progressive confinement within channels through both differential accretion and erosion. This paper extends that exciting result to describe the, again autogenic means by which flows can erode just beyond the channel mouths thereby prograding the channel system.

Taken together, these two papers for first time address the previously unconstrained speculation that has accompanied descriptions of observed relict deep-sea morphologies. Included in this speculation is a recent spate of papers that have built on each other citing the (still unproven) importance of assumed hydraulic jumps at these channel lobe transition zones (CLTZ). This new work, in contrast, demonstrates that there is no need for this inference and that a surprisingly non-intuitive, and previously unsuspected, mechanism can explain much of the observed morphology.

An important aspect is that these tank-scale experiments allow one to view the basal boundary layer which has never adequately been measured in the few field-scale current profiles. This paper presents clear evidence of a drop in the velocity maximum with a corresponding increase in the local bed shear stress. Having multiple closely spaced UVPS through the CLTZ, the onset and temporal and spatial evolution of this critical period of erosion is seen for the first time.

In the discussion they cite the lateral pressure gradients due to the loss of confinement as the cause of this flow “relaxation” (wouldn’t be my favourite term for this new mechanism, but not inaccurate). As they don’t have the UVPs spaced laterally, they don’t see it in this instance, but they refer to the 2016 paper again which does have exactly that geometry. This contribution is clearly described and I don’t thus see a need to recommend changes in that. I have a few questions, concerning their discussion which they might be able to address (in the limited # of words allowed):

We want to thank the reviewer for his kind words and for taking the time to provide this review.

- They don’t mention measured suspended sediment concentration (SSC) in the resulting flow at any point – I don’t think the UVP scattering can provide this. In the previous paper they merely assumed that it didn’t change from the incoming suspension (17% in this case). Do they assume the same or suspect a loss or gain in SSC?

The reviewer is correct that we could not measure the sediment concentration directly. We assume that changes in sediment concentration are small. This assumption is validated in part by the fact that only ~2% of the initial sediment volume is deposited on the slope. Thus, the changes to the amount of sediment in transport are minor. To prevent confusion, we now explicitly stated this in the Materials and Method section.

Changes applied to the manuscript:

Additional text, line 292-294:

“...with C as the sediment concentration at the inlet. Using the inlet concentration throughout the domain is validated by the fact that only ~2% of the initial sediment volume is deposited on the slope. Thus, changes to the amount of sediment in transport due to deposition are minor.”

- Does the UVP scattering provide any indication as to whether the vertical SSC profile similarly collapse with the velocity maximum?

The reviewer is right that this additional data would be of value. However, it is far from straight forward to calculate SSC from backscatter, especially with respect to the high sediment concentration used in the experiments (Thorne & Hanes, 2002; Thorne & Hurter, 2014). Attempts at such inversions are still in the development stage of research in our laboratory, and the results are not yet reliable. We have therefore decided not to use these inversions for the present manuscript.

References:

Thorne, P. D. & Hanes, D. M. 2002. A review of acoustic measurement of small-scale sediment processes. *Cont. Shelf Res.* 22, 603–632.
[https://doi.org/10.1016/S0278-4343\(01\)00101-7](https://doi.org/10.1016/S0278-4343(01)00101-7)

Thorne, P. D. & Hurter, D. 2014. An overview on the use of backscattered sound for measuring suspended particle size and concentration profiles in non-cohesive inorganic sediment transport studies. *Cont. Shelf Res.* 73, 97–118. <https://doi.org/10.1016/j.csr.2013.10.017>

Fig 4d shows the flow “thinning” but the plotted thickness of the flow is based on where the velocity is $\frac{1}{2}$ that of the peak – not the thickness of the suspended sediment layer – could the UVP confirm that the height to clear water also shrank?

We thank the reviewer for pointing out the ambivalent interpretation of “thinning”. Unfortunately, UVPs cannot detect the interface between clear water and the top of a turbidity current. Thicknesses thus have to be defined based on the architecture of the velocity profile, which is fundamentally satisfactory. What we observe in our experiments is a thinning of the velocity profile, which can be quantified easily with the elevation of the maximum velocity or the 50th percentile velocity in the mixing layer. This was not stated clearly enough in the manuscript. We have now rephrased the following sentences in the manuscript to accurately reflect which evidence we see in the UVP data for flow thinning.

Changes applied to the manuscript:

Line 119-121:

“...spread ~~and thinned~~ upon leaving the confinement. ~~This spreading lowered the elevation of the velocity maximum, and resulting~~ ~~resulted~~ in an increased basal ~~shearing~~ shear stress, which enhanced the ~~and-erosional~~ potential of the flow.”

Line 127-128:

“However, due to the lowering of the velocity maximum ~~thinning of the flow~~,...”

Line 130:

~~“The This~~ increased shear velocity ~~upon thinning of the flow is~~ was responsible for...”

Line 155-157:

“Our results indicate that this deformation consists of a lowering of the velocity maximum associated with ~~manifests itself through the vertical thinning and~~ lateral spreading of the flow field.”

Line 180-181:

~~“...as the flow thickness did not increase is thinning~~ upon leaving the confinement (Fig. 4d), ...”

Line 259-260:

~~“...spreads laterally spreads,~~ bringing the velocity maximum closer to the bed ~~thins,...~~”

Line 270:

~~“..., lateral spreading and thinning triggers a lowering the velocity maximum.”~~

- They are careful not to describe this flow as either super or sub-critical. I think this is wise given the popular bandwagon of use/misuse of these terms. They do state however that, by virtue of there being no hydraulic jump, the flow state doesn't change – would they be confident to say that it remain supercritical throughout?

We have strong indications that flows remain supercritical. By using the input sediment concentration of 17 vol% we calculate supercritical Froude throughout the experiments. However, as stated by the reviewer, we do not include the Froude number here as the relevance of a depth averaged Froude number remains speculative (see Waltham 2004; Sumner et al. 2013).

References:

Sumner, E.J., Peakall, J., Parsons, D.R., Wynn, R.B., Darby, S.E., Dorrell, R.M., McPhail, S.D., Perrett, J., Webb, A., White, D., 2013. First direct measurements of hydraulic jumps in an active submarine density current. *Geophys. Res. Lett.* 40, 5904–5908. <https://doi.org/10.1002/2013GL057862>

Waltham, D., 2004. Flow Transformations in Particulate Gravity Currents. *J. Sediment. Res.* 74, 129–134. <https://doi.org/10.1306/062303740129>

- They mention a topographic “noise” in the incision depth that was in their laser-scanned post-flow drained topography. Would they comment on whether this roughness had any periodicity that might be indicative of scour spacing or bedforms? Would such larger roughness elements even have any scaleable meaning?

We thank the reviewer for pointing this out. We do not see any spaced scours or bedforms in the channel thalweg. However, the channel thalweg in the reference experiment is incised deeper in the axis of the channel than at the flanks. By averaging over a 30 cm wide strip we calculate an average value for the incision in the channel.

We agree with the reviewer that the word “noise” was confusing in this context and therefore removed it.

Changes applied to the manuscript:

Line 352:

“...to remove ~~“noise” associated with~~ local variations in incision depth...”

- This particular flow exhibited the all-important short distance downstream of the loss of confinement where the shear velocity of the unconfined flow was greater than the confined flow (Fig 4 e). This occurs even though the flow is already significantly slower than the confined one (Fig 4c). As far as I can see this experiment was only run once (each for confined and unconfined). (Is this true?)

A total of eight runs have been conducted for these experiments. These include four test-runs, which were not of scientific value due to problems with malfunctioning equipment and the experimental setup.

Two of the successful experiments are presented here and the remaining two have not previously been a part of this paper. Although flow relaxation was also visible in the two neglected experiments, we did not use these experiments as the channel dimensions were too small for the modelled turbidity current (see last comment). However, if the editor or the reviewer feels that including these experiments is of value, we are happy to present the neglected experiments in the supplementary material.

Would they speculate whether, with a slight change in either the input slope or SSC (but still meeting their Shield scaling) that this zone of enhanced erosion would get longer or shorter?

We agree with the reviewer that it would be interesting to know how the flow relaxation and the area of erosion varies with changing input conditions. However, we feel that the main scope of this paper is to introduce the flow relaxation and the general concept of this new flow mechanism. Variation of this mechanism or the area of erosion by changing input conditions would require follow-up experiments in a different experimental set-up.

- While I think the concept is clearly demonstrated with this (singular?) tank experiment, it is just a result of a single instance of the flow. The flow matched the preexisting topography in the sense that it was completely constrained by the “levees” upstream and had sufficient momentum to not decelerate too rapidly. Any real CLTZ will, however, receive a wide range of flow competences. Some will exhibit this mechanism, others may die at this transition, and yet others be already out of equilibrium with the preexisting channel (already overwhelming the relief). Would they be prepared to comment on how varying incoming flow competencies might alter the presence and/or length scale over which this relaxation extends?

We thank the reviewer for this suggestion. We now discuss the relation of the incoming flow size with possible occurrence of flow relaxation and associated channel propagation in the manuscript.

Changes applied to the manuscript:

Additional text: Line 240-247:

“Channel propagation in the experiment resulted in a more gradual and natural loss of confinement, which still triggered flow relaxation and erosion. Additionally, channel propagation in natural systems will depend on the size of the turbidity currents with respect to the channel dimensions. If turbidity currents are too small to reach the end of the channel, flow relaxation cannot occur and the channel will not propagate, as seen for the Southern Channel in the Squamish Delta between 2004 and 2006 ^{26,44}. On the other hand, if a turbidity current is too large for the channel, then the flow will overflow the channel, thereby reducing its size until the flow matches the channel dimensions ⁴⁵, in which case flow relaxation will occur.”

These questions are perhaps more for my curiosity. I do not insist that they be answered. I believe the paper stands already as acceptable by Nature Comms. I look forward to it coming out.

John HC

REVIEWERS' COMMENTS:

Reviewer #1 (Remarks to the Author):

I would like to congratulate the authors for their work compiling the reviewers' comments and giving solid response to them. The authors addressed and satisfactorily responded all observations in my review. I have no further comments on the revised manuscript, and based on it, I suggest that the editor accepts the revised manuscript for publication. I look forward to reading the published work soon,
Maria Azpiroz-Zabala

Reviewer #3 (Remarks to the Author):

Thoughts on response from Pohl et al to reviewers' comments:

I believe they have addressed all reviewer concerns (most of which were semantics). Concerning my questions, I can confirm that I'm satisfied with their responses. Specifically they mention that there are other experimental runs that they hadn't reported on. If they would rather not include them (not critical to the key point of the paper), I'm satisfied with that.
At this point I feel this paper should be fully acceptable to Nature Comm.
Line 463 – prove-reading should be proof-reading
JHC

Reviewer #1 (Remarks to the Author):

I would like to congratulate the authors for their work compiling the reviewers' comments and giving solid response to them. The authors addressed and satisfactorily responded all observations in my review. I have no further comments on the revised manuscript, and based on it, I suggest that the editor accepts the revised manuscript for publication. I look forward to reading the published work soon,
Maria Azpiroz-Zabala

We thank the reviewer for their kind words and for the time she spend on reviewing our manuscript.

Reviewer #3 (Remarks to the Author):

I believe they have addressed all reviewer concerns (most of which were semantics). Concerning my questions, I can confirm that I'm satisfied with their responses. Specifically they mention that there are other experimental runs that they hadn't reported on. If they would rather not include them (not critical to the key point of the paper), I'm satisfied with that.

At this point I feel this paper should be fully acceptable to Nature Comm.

Line 463 – prove-reading should be proof-reading

JHC

We want to thank the reviewer for their kind words and for taking the time to provide this review. We corrected the typo in line 463.